# Combining Latent Space and Structured Kernels for Bayesian Optimization over Combinatorial Spaces

**Aryan Deshwal**
School of EECS
Washington State University
aryan.deshwal@wsu.edu

**Janardhan Rao Doppa**
School of EECS
Washington State University
jana.doppa@wsu.edu

## Abstract

We consider the problem of optimizing combinatorial spaces (e.g., sequences, trees, and graphs) using expensive black-box function evaluations. For example, optimizing molecules for drug design using physical lab experiments. Bayesian optimization (BO) is an efficient framework for solving such problems by intelligently selecting the inputs with high utility guided by a learned surrogate model. A recent BO approach for combinatorial spaces is through a reduction to BO over continuous spaces by learning a latent representation of structures using deep generative models (DGMs). The selected input from the continuous space is decoded into a discrete structure for performing function evaluation. However, the surrogate model over the latent space only uses the information learned by the DGM, which may not have the desired inductive bias to approximate the target black-box function. To overcome this drawback, this paper proposes a principled approach referred as LADDER. The key idea is to define a novel structure-coupled kernel that explicitly integrates the structural information from decoded structures with the learned latent space representation for better surrogate modeling. Our experiments on real-world benchmarks show that LADDER significantly improves over the BO over latent space method, and performs better or similar to state-of-the-art methods.

## 1 Introduction

A huge range of science and engineering applications involve optimizing combinatorial spaces (e.g., sequences, trees, graphs) using *expensive* black-box function evaluations [18, 70, 17]. For example, in drug design application, each candidate structure is a molecule and evaluation involves performing an expensive physical lab experiment. Bayesian optimization (BO) [59, 20] is an effective framework for optimizing expensive black-box functions and has shown great success in practice [63, 74, 21, 6, 5]. The key idea is to learn a *cheap-to-evaluate* surrogate statistical model, e.g., Gaussian process (GP), from past function evaluations and employ it to select inputs with high-utility for evaluation. BO for combinatorial spaces is a relatively less-studied problem with many challenges in the *small-data* setting (number of function evaluations is small).

A recent approach to solve some of these challenges is through a reduction to BO over continuous spaces, which we refer as *BO over latent space* [26, 68]. This method relies on two ideas. First, we learn a latent space representation from a database of unsupervised structures (i.e., function evaluations are not available) using a encoder-decoder style deep generative model (DGM) [37, 33]. Second, we build a surrogate model over this latent space to perform BO. Each selected input from the latent space is decoded into a structured object for performing function evaluation. BO over latent space has shown good success when the number of function evaluations is large [26, 68, 19, 34]. However, we conjecture that this approach may not be effective in the small-data settings for the following reasons. **1)** The surrogate model over the latent space only uses the information learned by the DGM, which may not have the desired inductive bias to approximate the target black-box

function. **2)** The learned surrogate model may not generalize well beyond the training instances from the latent space. Indeed, our experiments on real-world problems demonstrate the ineffectiveness of this Naïve surrogate model over the latent space.

In this paper, we propose a novel approach referred as LADDER to overcome the above drawbacks for the small-data setting. We name it LADDER because it acts as a ladder in connecting the rich structural information of structures in the combinatorial space with their corresponding latent space representations. Intuitively, it improve surrogate modeling by combining the best of both representations. We provide a principled Gaussian process based method to integrate the latent space representation with the structural information from decoded structures using a novel *structure-coupled kernel*. The key insight is to extend a kernel over the latent space (with its hyper-parameters estimated on the evaluated points) to non-evaluated points in the space by utilizing the features of structured kernels defined over combinatorial spaces (e.g., string kernels and graph kernels). LADDER has multiple advantages. First, we can leverage a large body of research on kernels over structured data. Second, allows the use of deep generative models (DGMs) to learn latent space as a plug-and-play technology. This means advances in DGMs will directly improve the effectiveness of LADDER. Our experiments on real-world benchmarks show that LADDER performs better or similar to state-of-the-art methods, and significantly better than the Naïve latent space BO approach in our problem setting. We also empirically demonstrate that superiority of LADDER's performance is due to better surrogate model resulting from the proposed method to combine representations.

**Contributions.** The key contribution of this paper is the development and evaluation of the LADDER approach to perform BO over combinatorial spaces in the *small-data setting*. Specific list includes:

- Identifying the key reasons for the ineffectiveness of the Naïve latent space BO method in the small-data setting and providing empirical evidence on real-world problems.

- Development of a principled Gaussian process approach for improved surrogate modeling by combining the structural information from decoded structures with the learned latent space representation using a novel structure-coupled kernel.

- Experiments on real-world benchmarks to show the efficacy of LADDER over prior methods in our problem setting. The code and data are available on the GitHub repository link `https://github.com/aryandeshwal/LADDER`.

## 2 Problem Setup and Background

Let $\mathcal{X}$ be a space of combinatorial structures (e.g., sequences, trees, and graphs). We assume the availability of a black-box objective function $f : \mathcal{X} \mapsto \mathbb{R}$ defined over the combinatorial space $\mathcal{X}$. Evaluating each candidate structure $\mathbf{x} \in \mathcal{X}$ using function $f$ (also called an experiment) is *expensive* in terms of the resources consumed and produces an output $y = f(\mathbf{x})$. For example, in the drug design application, each $\mathbf{x} \in \mathcal{X}$ is a molecule, and $f(\mathbf{x})$ corresponds to running a physical lab experiment. Our overall goal is to find a structure $\mathbf{x} \in \mathcal{X}$ that approximately optimizes $f$ by minimizing the number of experiments and observing their outcomes.

We are also provided with a database of *unsupervised* structures $\mathcal{X}_u \subset \mathcal{X}$. Unsupervised means that we do not know the function evaluations $f(x)$ for structures in $\mathcal{X}_u$. This assumption is satisfied by many scientific applications including chemical design and material design. We assume the availability of a *latent space* $\mathcal{Z}$ learned from unsupervised structures $\mathcal{X}_u$ using a encoder-decoder style deep generative model, e.g., variational autoencoders (VAEs) for structured data such as junction tree VAE [33] and grammer VAE [37]. Formally, the encoder denoted by $\Upsilon$ embeds a given combinatorial structure $\mathbf{x} \in \mathcal{X}$ into a point in the latent space $\mathbf{z} \in \mathbb{R}^d = \Upsilon(x)$ where $d$ is the number of dimensions of the latent space $\mathcal{Z}$ and the decoder denoted by $\Phi$ converts a given point from latent space $\mathbf{z}' \in \mathcal{Z}$ into a structured object $\mathbf{x}' \in \mathcal{X} = \Phi(\mathbf{z}')$. Encoder $\Upsilon$ and decoder $\Phi$ are typically realized by neural networks.

**Bayesian optimization.** To solve optimization problems using black-box evaluations of expensive objective functions, Bayesian optimization (BO) is an efficient framework. BO algorithms select inputs with high utility guided by a learned surrogate statistical model from past observations. The three key elements of BO are as follows: **1)** Surrogate model of the black-box function $f(\mathbf{x})$, e.g., Gaussian process; **2)** Acquisition function to score the utility of evaluating candidate inputs using the surrogate model, e.g., expected improvement; and **3)** Acquisition function optimizer to select the input with maximum utility. In each BO iteration, we select one input for function evaluation

and update the statistical model based on the new training example. BO has shown great success for continuous spaces, but BO over combinatorial spaces is a relatively less-studied problem space.

**Naïve latent space BO.** This approach reduces the problem of BO over combinatorial space $\mathcal{X}$ to BO over the latent space $\mathcal{Z}$, which is continuous. We construct surrogate model $\mathcal{M}_{\mathcal{Z}}$ over the latent space $\mathcal{Z}$ using training points selected from $\mathcal{Z}$ for evaluation. Each selected input $\mathbf{z} \in \mathcal{Z}$ is decoded into a structured object using the decoder $\Phi$. We perform function evaluation using the decoded structure $\Phi(\mathbf{z})$ and the surrogate model $\mathcal{M}_{\mathcal{Z}}$ is updated using the new training example $(\mathbf{z}, f(\Phi(\mathbf{z})))$. This reduction approach allows us to leverage prior research on BO for continuous spaces and has shown success when the the number of function evaluations is large [26, 68, 19, 34]. In this paper, we study the *small data* setting (number of function evaluations is small) by addressing the drawbacks of this Naïve latent space BO approach in a principled manner.

## 3 LADDER: Latent Space BO guided by Decoded Structures

In this section, we first discuss the challenges with the Naïve latent space BO approach. Next, we describe our proposed LADDER approach with a focus on the novel surrogate model by combining kernels over structured data and latent space representation, which is our key technical contribution.

### 3.1 Challenges with the Naïve latent space BO approach

As mentioned above, the Naïve latent space BO approach builds a surrogate model over the latent space using kernels for continuous spaces (e.g., Matern or Squared Exponential kernel) and performs acquisition function optimization in the latent space using optimizers for continuous spaces (e.g., gradient-based methods) to select point $\mathbf{z} \in \mathcal{Z}$ for evaluation. However, the expensive objective function $f(\mathbf{x})$ is defined over the space of combinatorial structures $\mathcal{X}$ and *not* the latent space $\mathcal{Z}$. Therefore, we need to decode this point $\mathbf{z}$ using the decoder $\Phi$ to get the corresponding combinatorial structure $\mathbf{x}=\Phi(z)$ for function evaluation $f(\mathbf{x})$. All the existing work on BO over latent space do not account for this decoding process. As a consequence, we need to deal with two inter-related challenges, which are especially significant for small-data settings.

- **Challenge #1:** The kernel over the latent space only uses the information learned by the deep generative model. It doesn't explicitly incorporate information about the decoded structure. This means the corresponding Gaussian process surrogate model may not have the desired inductive bias to approximate the black-box objective function. Therefore, we are not able to leverage this potentially useful inductive bias and rich structural information available in decoded structures.
- **Challenge #2:** The surrogate statistical model itself might not generalize well beyond the training examples from the latent space (set of points in the latent space and their corresponding function evaluations). This is especially true in the small-data setting for latent spaces learned using DGMs in real-world scientific applications, where the number of dimensions of latent space can be large when compared to the standard BO setup.

Indeed, we provide empirical evidence to demonstrate these challenges and our key hypothesis in Figure 2. We show that by incorporating the rich structural information from the decoded output, we can address both these challenges to improve the overall BO performance.

### 3.2 Overview of LADDER algorithm and key advantages

LADDER is an instantiation of the latent space BO framework that employs a novel surrogate statistical model to address the two challenges with the Naïve method. The surrogate model is a Gaussian process that combines the complementary strengths of the latent space representation with the rich structural information from decoded outputs using structured kernels (e.g., string kernels and graph kernels). The key idea is to define a *structure-coupled kernel* that extends the continuous kernel on the evaluated points in the latent space to unknown points using the rich information from structured kernels. We employ expected improvement (EI) as the acquisition function. For optimizing the acquisition function to select high utility inputs from the latent space, we employ evolutionary search due to its recent successes [39] including policy search in high-dimensional spaces [55].

Figure 1 shows a high-level illustration of LADDER and Algorithm 1 provides the complete pseudo-code. We use a small set of initial training data in the form of points in the latent space and their

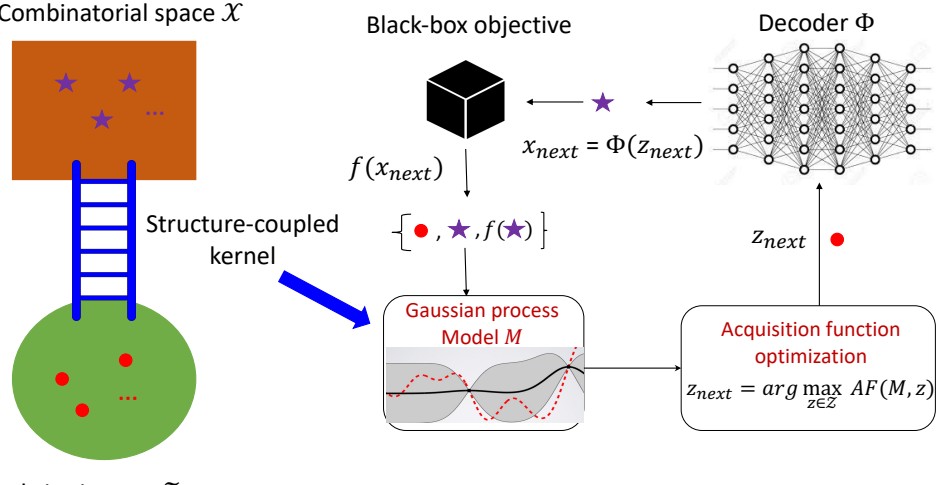

Combinatorial space $\mathcal{X}$

Black-box objective

Decoder $\Phi$

$f(x_{next})$

$x_{next} = \Phi(z_{next})$

Structure-coupled kernel

$z_{next}$

Gaussian process Model $M$

Acquisition function optimization

$z_{next} = arg \max_{z \in \mathcal{Z}} AF(M, z)$

Latent space $\mathcal{Z}$

Figure 1: High-level conceptual illustration of our proposed LADDER approach, which acts as a "ladder" in connecting the rich structural information of each structure in the combinatorial space with its corresponding latent space representation. Structure-coupled kernel is the key element that enables this connection to build an effective Gaussian process based surrogate model.

corresponding function evaluations to bootstrap the GP based surrogate model using the structure-coupled kernel. In each iteration $t$, we optimize the acquisition function to select a point $\mathbf{z}_t$ from the latent space $\mathcal{Z}$ for evaluation. The corresponding decoded structure $\mathbf{x}_t = \Phi(\mathbf{z}_t)$ is evaluated to measure the outcome $f(\mathbf{x}_t)$. The GP model with structure-coupled kernel is updated using the new 3-tuple training example $\{\mathbf{z}_t, \mathbf{x}_t, f(\mathbf{x}_t)\}$. We repeat these sequence of steps until covergence or maximum query budget and then return the best uncovered combinatorial structure $\hat{\mathbf{x}} \in \mathcal{X}$ as the output.

**Advantages of LADDER.** Some of the key advantages of our proposed approach are listed below.

- We are allowed to employ any existing trained deep generative model for structured data in a *plug-and-play* manner with LADDER. Therefore, any advances in the latent-space generative modeling technology will directly improve the overall BO performance.

- LADDER is a generic approach that is applicable to any combinatorial space of structures (e.g., sequences, trees, graphs, sets, and permutations). This method just requires an appropriate structured kernel over the given combinatorial space. Therefore, we can leverage a large body of research on generic kernels over structured data (e.g., string kernels and graph kernels) and hand-designed kernels based on domain knowledge for specific applications. For example, in our experiments, we employ sub-sequence string kernel (generic) and fingerprint kernel (domain-specific) to concretely instantiate the LADDER approach for strings and molecules respectively to demonstrate its flexibility.

- Combines the complementary strengths of latent space representations and structured kernels in a principled manner to create highly-effective surrogate statistical models.

### 3.3 Novel surrogate statistical model via structure-coupled kernel

We consider *Gaussian process (GP)* [52] as the surrogate model of the expensive black-box objective function $f(\mathbf{x} \in \mathcal{X})$. GPs are known to have excellent statistical properties including principled uncertainty quantification, which is critical for the effectiveness of BO algorithms. A GP model defines a prior distribution on functions which is entirely characterized by a kernel defined over a pair of inputs. Most of the existing work on latent space BO employs a standard continuous space kernel represented by $l(\mathbf{z_i} \in \mathcal{Z}, \mathbf{z_j} \in \mathcal{Z})$, e.g., Radial Basis Function (RBF) / Gaussian and Matern kernels, over points in the latent space to create the GP model. However, this surrogate model is

**Algorithm 1** Latent Space Bayesian Optimization guided by Decoded Structures (LADDER)

1: **Input:** Objective function $f(\mathbf{x})$, Encoder ($\Upsilon$) - Decoder ($\Phi$) style model for latent space $\mathcal{Z}$, Kernel for latent space $l(\mathbf{z_i} \in \mathcal{Z}, \mathbf{z_j} \in \mathcal{Z})$, Kernel for combinatorial space $k(\mathbf{x_i} \in \mathcal{X}, \mathbf{x_j} \in \mathcal{X})$

2: Initialize dataset $\mathcal{D}_0$ by evaluating few random points: $\mathcal{D}_0 \leftarrow \{\mathbf{Z}_0, \mathbf{X}_0, f(\mathbf{X}_0)\}; t \leftarrow 0$
   // slight abuse of notation here since $\mathbf{Z}_0$ is the set of initial points from latent space $\mathcal{Z}$ and $\mathbf{X}_0$ is the set of corresponding decoded structures from $\mathcal{X}$ with function evaluations $f(\mathbf{X}_0)$

3: **repeat**

4:     Learn Gaussian process model on the dataset $\mathcal{D}_t$ with the proposed kernel in Equation 4

5:     Optimize acquisition function over the latent space $\mathcal{Z}$ to find the next point $\mathbf{z}_t$ for evaluation

6:     Compute the decoded structure $\mathbf{x}_t$ for point $\mathbf{z}_t$ using decoder $\Phi$

7:     Evaluate the combinatorial structure $\mathbf{x}_t$ to get $f(\mathbf{x}_t)$

8:     Add new training triple: $\mathcal{D}_{t+1} \leftarrow \mathcal{D}_t \cup \{\mathbf{z}_t, \mathbf{x}_t, f(\mathbf{x}_t)\}$; increment the iteration $t \leftarrow t + 1$

9: **until** convergence or maximum iterations

10: **Output:** best uncovered structure $\hat{\mathbf{x}}$ and the corresponding function value $f(\hat{\mathbf{x}})$

---

highly-ineffective for small data settings, especially when the number of dimensions of latent space is large, which is the common case for deep generative models for scientific applications.

**Structure-coupled kernel.** We propose to utilize the rich structural information that is available from the decoded combinatorial structure $\mathbf{x}$ corresponding to each point $\mathbf{z}$ from the latent space $\mathcal{Z}$. The key idea behind our approach is to integrate the structure information from the decoded outputs with the learned representation of inputs from the latent space to achieve better surrogate modeling performance. To include this structure information in a principled manner within a GP model, we leverage the Generalized Nystrom extension idea [57, 69] to *extrapolate* the eigenfunctions of the kernel matrix over latent space ($\mathbf{L} = \{L_{ij} = l(\mathbf{z_i}, \mathbf{z_j}) | \mathbf{z_i}, \mathbf{z_j} \in \mathcal{Z}\}$) with basis functions from a kernel $k(\mathbf{x}_i \in \mathcal{X}, \mathbf{x}_j \in \mathcal{X})$ defined over the decoded combinatorial structures.

Without loss of generality, let $m$ be the number of evaluated inputs from the latent space, which are denoted as $\mathbf{Z} = \{\mathbf{z_1}, \mathbf{z_2}, \cdots, \mathbf{z_m}\}$. For example, these are the points accumulated after $m$ iterations of the latent space BO approach. Let the corresponding set of decoded structures be $\mathbf{X} = \{\mathbf{x_1} = \Phi(\mathbf{z_1}), \mathbf{x_2} = \Phi(\mathbf{z_2}), \cdots, \mathbf{x_m} = \Phi(\mathbf{z_m})\}$ with their function evaluations $\{f(\mathbf{x_1}), f(\mathbf{x_2}), \cdots, f(\mathbf{x_m})\}$. Given kernels $l : \mathbf{z} \times \mathbf{z} \to \Re$ and $k : \mathbf{x} \times \mathbf{x} \to \Re$ with $\mathbf{L}$ and $\mathbf{K}$ representing their corresponding kernel matrices, Generalized Nystrom extension generates an $m$-dimensional feature vector $\xi(\mathbf{z})$ for a point $\mathbf{z}$ in the latent space. The $i$th component of $\xi(\mathbf{z})$ is given as follows:

$$\xi_i(\mathbf{z}) = {\mathbf{k_z}}^T \mathbf{K}^{-1} v_i \quad i \in \{1, 2, \cdots, m\} \tag{1}$$

where $\mathbf{k_z}$ is an $m$-dimensional vector evaluated between $\mathbf{x}$ (decoded output of latent space input $\mathbf{z}$) and other combinatorial structures from $\mathbf{X}$:

$$\mathbf{k_z} = [k(\Phi(\mathbf{z}), \Phi(\mathbf{z_1})), \cdots, k(\Phi(\mathbf{z}), \Phi(\mathbf{z_m}))] = [k(\mathbf{x}, \mathbf{x_1}), \cdots, k(\mathbf{x}, \mathbf{x_m})] \tag{2}$$

$\mathbf{K}$ is an $m \times m$ kernel matrix for combinatorial structures in the set $\mathbf{X}$, i.e., $K_{ij} = k(\Phi(\mathbf{z_i}), \Phi(\mathbf{z_j})) = k(\mathbf{x_i}, \mathbf{x_j})$, and $v_i$ is the eigenvalue-scaled eigenvector of the kernel matrix $\mathbf{L}$ defined over the latent space inputs in the set $\mathbf{Z}$, i.e. $V = [v_1, v_2, \cdots v_m] = U\Sigma^{1/2}$, where $U$ and $\Sigma$ are eigenvectors and eigenvalues of $\mathbf{L}$ respectively.

The reader should note that the term $k(\Phi(\mathbf{z_i}), \Phi(\mathbf{z_j}))$ in (2) means that the structured kernel is applied to the decoded structures $\mathbf{x_i} = \Phi(\mathbf{z_i})$ and $\mathbf{x_j} = \Phi(\mathbf{z_j})$ of latent space inputs $\mathbf{z_i}$ and $\mathbf{z_j}$ respectively. This is a key term in the above expression because it integrates each input from latent space with its decoded structure in the new feature map $\xi(\mathbf{z})$. For any two inputs $\mathbf{z}$ and $\mathbf{z}'$ in the latent space, the resulting structure-coupled kernel denoted as $c(\mathbf{z}, \mathbf{z}'))$ is defined as the dot product of their corresponding feature vectors $\xi(\mathbf{z})$ and $\xi(\mathbf{z}')$:

$$c(\mathbf{z}, \mathbf{z}') = \xi(\mathbf{z})^{\mathbf{T}} \xi(\mathbf{z}') \tag{3}$$

$$c(\mathbf{z}, \mathbf{z}') = {\mathbf{k_z}}^T \mathbf{K}^{-1} \mathbf{L} \mathbf{K}^{-1} \mathbf{k_{z'}} \tag{4}$$

Intuitively, by this construction, we are extending the kernel over the latent space $l$ on the evaluated points $\mathbf{Z}$ to non-evaluated points in the latent space by utilizing the rich structural features from kernel $k$ defined over combinatorial spaces. For training points (candidate inputs from latent space

along with their function evaluations), the resulting kernel matrix will be $\mathbf{L}$ since Equation 4 becomes $\mathbf{K}\mathbf{K}^{-1}\mathbf{L}\mathbf{K}^{-1}\mathbf{K} = \mathbf{L}$. For latent space points not in the training set, the structured kernel $k$ acts like a smooth extrapolating kernel. It can also be seen by interpreting the Equation 4 through the definition of a kernel in terms of the empirical kernel map [1](with respect to the decoded structured outputs) endowed with a dot product induced by the positive [2] definite matrix $\mathbf{K}^{-1}\mathbf{L}\mathbf{K}^{-1}$, i.e.,

$$c(\mathbf{z}, \mathbf{z}') = <\mathbf{k_z}, \mathbf{k_{z'}}>_{\mathbf{K}^{-1}\mathbf{L}\mathbf{K}^{-1}} = <\mathbf{k_z}, \mathbf{K}^{-1}\mathbf{L}\mathbf{K}^{-1}\mathbf{k_{z'}}> \tag{5}$$

Importantly, the above general construction of structure-coupled kernel allows us to leverage extensive research on kernel methods for highly-structured data, which try to exploit structural features of combinatorial objects. For example, string kernels [38] count the number of common sub-strings in string inputs, fingerprint kernels [51] capture neighborhood-aggregated properties of molecules, and features such as number of random walks and shortest paths are utilized by graph kernels [7].

## 4 Experiments and Results

In this section, we empirically evaluate the effectiveness of the proposed LADDER approach on real-world benchmarks, and perform comparison with baseline methods.

### 4.1 Real-world benchmarks

We employ two widely used real-world benchmarks for combinatorial Bayesian optimization.

**Arithmetic expressions optimization.** In this benchmark, the goal is to search in the space of uni-variate arithmetic expressions (generated from a given grammar) to find the best expression that fits a given target expression [37]. As described in [37], the latent space model is trained on 100K randomly generated expressions from the following grammar:

```
S → S '+' T | S '*' T | S '/' T | T
T → '(' S ')' | 'sin(' S ')' | 'exp(' S ')'
T → 'v' | '1' | '2' | '3'
```

We follow the same setup as discussed in the state-of-the-art paper for this benchmark [68]. We consider the log mean-squared error between an expression $\mathbf{x}$ and the target expression $\mathbf{x}^* = 1/3 \cdot v \cdot \sin(v^2)$ (computed over 1000 evenly-spaced values of $v$ in the interval $[-10, 10]$) as the objective function which should be minimized.

**Chemical design optimization.** This benchmark considers finding molecules with best drug-like properties [37] and is similar in prototype for many scientific applications. Specifically, the goal is to maximize the water-octanol partition coefficient (logP) over the space of molecules. The latent space model is trained on the Zinc molecule dataset of 250K molecules. For consistency purposes, all our results are shown as minimization obtained by taking a negation of the logP objective.

### 4.2 Experimental setup

**Configuration of algorithms.** The BO part of the source code is written using the popular GpyTorch [22] and BoTorch [2] libraries for all BO methods including LADDER. We employ ARD (automatic relevance determination) Matern kernel for the latent space inputs in all our experiments. Matern kernel is commonly advocated as a better choice than RBF kernel for BO algorithms since the sample functions from the latter are impractically smooth [63]. Hyperparameters of Gaussian process models are fitted by marginal likelihood maximization after every BO iteration. We employed Junction tree VAE [33] and Grammar VAE [37] as the latent-space model for chemical design and arithmetic expression optimization benchmarks respectively. Both pretrained encoder-decoder models are taken from the source code provided by the authors' of [68] [3]. We employed expected improvement (EI) as the acquisition function for all the BO methods. All experiments are performed on a machine with the following configuration: Intel(R) Core(TM) i9-7960X CPU @ 2.80GHz with 128 GB RAM.

---

[1]We refer to empirical kernel map as commonly defined in [56] (Definition 3)

[2]Positive definiteness of $\mathbf{K}^{-1}\mathbf{L}\mathbf{K}^{-1}$ can be easily seen as a consequence of the following three facts: i) $\mathbf{K}$ and $\mathbf{L}$, being kernel matrices, are positive definite; ii) inverse of a psd matrix is psd; and iii) $MNM$ is psd if $M$ and $N$ are two psd matrices.

[3]https://github.com/cambridge-mlg/weighted-retraining/

**LADDER instantiations.** In addition to the encoder-decoder style latent space model, which is same as the Naïve latent space BO (LSBO), LADDER also requires an appropriate structured kernel for the given combinatorial space. To demonstrate the generality of our proposed approach, we instantiate LADDER with two different kernels for our two optimization benchmarks. We employed sub-sequence string kernel for the arithmetic expressions task and fingerprints kernel for the chemical design task. We briefly describe both kernels below.

- *Sub-sequence string kernel.* This kernel captures the similarity between two strings by counting the number of matching substrings, where the substrings can be non-contiguous [38, 10]. Following the notation in [42], given an alphabet $\Pi$, the kernel between two strings $s_1$ and $s_2$ is given as: $k(s_1, s_2) = \sum_{u \in \Pi^n} \rho(s_1)\rho(s_2)$ where $\rho(s) = \lambda_m^{|u|} \cdot \sum_{1 < i_1 < \cdots < i_{|u|} < |s|} \left( \lambda_g^{i_{|u|} - i_1} \mathbf{1}_u(s_{i_1}, \cdots, s_{i_{|u|}}) \right)$ for any string $s$ where $\lambda_g$ and $\lambda_m$ are gap decay and match decay hyper-parameters. Since arithmetic expressions are naturally represented as strings, we use this kernel for the arithmetic expressions task.

- *Fingerprints kernel.* There is a large body of work in the chemical informatics literature for designing structural features for molecular inputs. These hand-engineered features by domain experts are commonly known as molecular fingerprints [41]. We employ Morgan fingerprints [53] which are high-dimensional binary features to capture different substructures in a molecule while being invariant to atom relabeling. Since combinatorial structures in the chemical design task are molecules, given two molecules $m_1$ and $m_2$, we consider the dot product of their Morgan fingerprints as the structured kernel for the chemical design task. Hence, we refer to this kernel as the Fingerprints kernel. We employed 2048-bit fingerprints with a bond radius of 3 [43].

In all our results and figures, the corresponding structured kernel for LADDER is denoted in the parenthesis, e.g., LADDER (String). We employed the evolutionary search algorithm CMA-ES [28] as the acquisition function optimizer for LADDER. The parameters of CMA-ES[4] were fixed with sigma = 0.2 and a population size of 50. The performance of CMA-ES was found to be highly robust to different choice of these parameters. We ran CMA-ES from 10 different starting inputs for 10 iterations each and picked the best optimizer found. As discussed later, we consider one instance of Naïve LSBO with the same configuration of CMA-ES optimizer for fair comparison and to test our key hypothesis that surrogate modeling within LADDER is better. We use 10 random points (uniformly picked from the dataset) to initialize the GP models.

**Evaluation metric.** We evaluate all methods on the best objective (log MSE for the arithmetic expressions task and logP for the chemical design task) value uncovered as a function of the number of experiments (expensive function evaluations). The method that finds high-performing structures with less number of function evaluations is considered better.

We ran each method on all benchmarks for 10 different runs with the same initialization (small number of randomly selected structures and function evaluations). We plot the mean and two times the standard error for all our experimental results.

### 4.3 Results and discussion

**Comparison of surrogate models.** Recall that the key hypothesis of this paper is that surrogate model employed by the Naïve LSBO approach (GP model with Matern kernel in our case) is ineffective and GP model with our proposed structure-coupled kernel for small-data settings. To test this hypothesis, we compare the quality of the model fit for these two GP models (Naïve LSBO and LADDER) by evaluating the mean absolute error (MAE) of their predictions on a testing set. To perform this experiment, we generate 50 (uniformly) random training sets of different sizes and evaluate the models on 20 random testing sets. The averaged MAE results over these 1000 training and testing set pairs are shown in Figure 2. We make following observations. **1)** The standard GP model on latent space denoted by Naïve LSBO shows some improvement in MAE for the arithmetic expressions task while the improvement is minimal for the chemical design task. **2)** Our proposed surrogate model, i.e., GP with structure-coupled kernel, denoted by LADDER has significantly lower MAE than Naïve LSBO on both the benchmarks and continuously decreases as a function

---

[4]https://github.com/CMA-ES/pycma

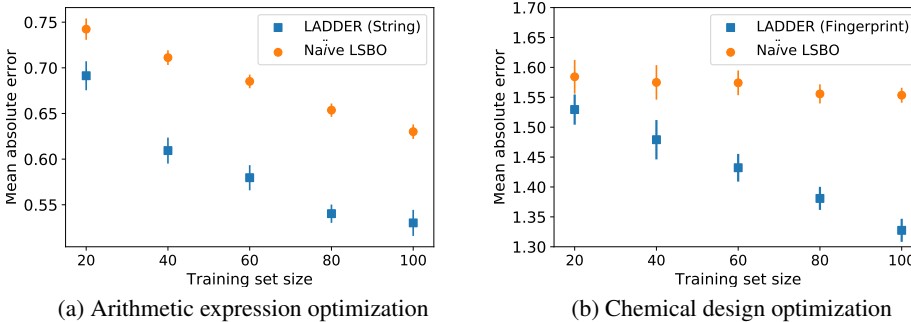

(a) Arithmetic expression optimization  (b) Chemical design optimization

Figure 2: Mean absolute error results comparing the quality of model fit for varying sizes of training sets with two models: GP model with Matern kernel (Naïve LSBO) and GP model with the proposed structure-coupled kernel (LADDER). Lower MAE values mean better surrogate model.

of the training set size. These strong results corroborate our key hypothesis and demonstrate the effectiveness of our proposed surrogate model with the structure-coupled kernel.

**Naïve latent space BO vs. LADDER.** A natural question based on the above results is whether improved surrogate modeling results in better overall BO performance. To answer this question, we compare the BO performance (best uncovered or incumbent objective value vs. number of function evaluations or iterations) of Naïve LSBO and LADDER. We consider two choices for acquisition function optimizer within the Naïve LSBO approach: zeroth order CMA based optimizer (same as LADDER) and second-order gradient based optimizer (L-BFGS). We make the following observations from the results shown in Figure 3. **1)** LADDER consistently uncovers significantly better structures than those from the Naïve LSBO approach on both benchmarks. This is a direct consequence of the better surrogate model of LADDER since the acquisition function optimizer is kept the same for both methods, i.e., CMA. **2)** L-BFGS based acquisition function optimizer slightly improves the BO performance of Naïve LSBO but still cannot match the performance of LADDER.

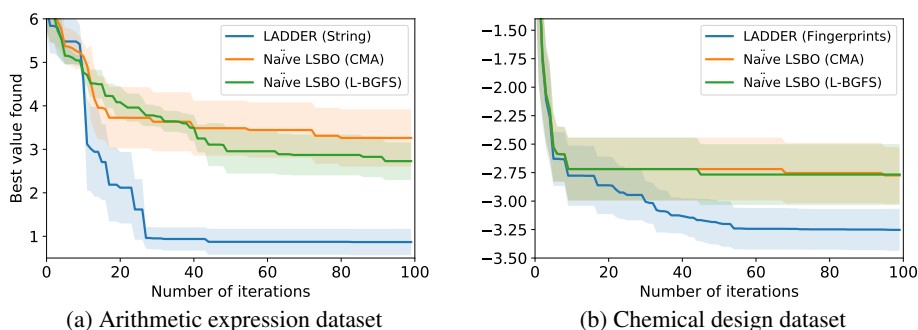

(a) Arithmetic expression dataset  (b) Chemical design dataset

Figure 3: Results comparing the BO performance of LADDER and Naïve latent space BO.

**LADDER vs. State-of-the-art.** To further analyze LADDER, we compare its performance with a set of state-of-the-art methods. We include *weighted retraining* approach which was recently proposed [68] to update the latent space model as the BO algorithm progresses (Naïve LSBO w/ retraining). We also consider design by adaptive sampling (DbAS) [8], the cross-entropy method with probability of improvement (CEM-PI) [54], the feedback VAE (FB-VAE) [27], and reward-weighted regression (RWR) [49]. We employ the publicly available implementation of these baseline methods. We make the following observations from the results shown in Figure 4. **1)** LADDER performs significantly better on the arithmetic expressions task and similar to the best method on the chemical design task. We verify the similar performance of LADDER on the chemical design task by performing a two-sided paired Wilcoxon test at 1% significance. The performance of LADDER and Naïve LSBO w/ retraining is statistically similar on the chemical design task (p-value = 0.0489). **2)** Retraining

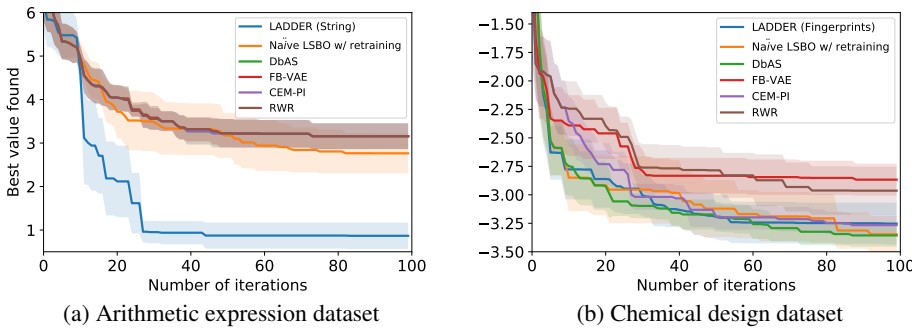

(a) Arithmetic expression dataset

(b) Chemical design dataset

Figure 4: Results comparing the BO performance of LADDER and different state-of-the-art methods.

the latent space model helps in improving the performance of Naïve LSBO on both tasks. **3)** DbAS and CEM-PI perform similar to Naïve LSBO w/ retraining since they are special case of retraining method as described in [68]. The discrepancy in LADDER's performance on the chemical design benchmark can be attributed to the low-flexibility of fingerprint kernel which doesn't include any hyper-parameters (as opposed to string kernel) to tune it for a specific dataset.

## 5 Related Work

In this section, we discuss the most closely related work for the problem setting studied in this paper.

**Deep generative models for structured data.** Scientific domains including chemistry and molecular biology involve highly structured data [58]. To address the challenges of structured data, there is a growing body of work on deep generative models (DGMs). Some notable examples include grammar VAE [37], junction tree VAE [33] and syntax-directed VAE [11]. This line of work is complementary to our work as advances in DGMs will automatically improve the BO performance of LADDER.

**Kernels over structured data.** There is a large body of research on generic kernels over structured data including strings [38, 10], trees [61], graphs [7], sets [9], and permutations [32]. LADDER allows us to leverage this prior research to perform sample-efficient BO over combinatorial spaces.

**BO over latent space.** The use of DGMs to learn a latent space and to perform BO in this continuous space was first started by Gomez-Bombarelli et al., for chemical design [26]. This led to a lot of followup work motivated by applications from multiple domains [40, 19, 34, 33]. A recent work by Tripp et al., [68] noted that this decoupled approach of training DGM and BO process can make the overall optimization problem hard; and proposed an approach referred as weighted retraining. The key idea is to periodically retrain the DGM by giving more weight to inputs with high function values. Our LADDER approach and weighted retraining have complementary strengths. For small-data settings, LADDER is much more effective than weighted retraining as demonstrated by our experiments.

**BO over combinatorial space.** We can categorize these methods based on the choice of surrogate model and search strategies for acquisition function optimization. Linear models [3], Gaussian processes with kernels over discrete structures [46, 15, 24, 36], and random forest [30] are considered for surrogate modeling. Search strategies include mathematical optimization [3, 13, 15], heuristic search methods [30, 46, 14], and combination thereof [16, 66, 44]. [31] provided a BO approach for tree structured spaces. LADDER is complementary to this line of work. First, it allows to combine the complementary strengths of two pieces of knowledge, namely, generic/domain-specific kernels over structured data and data-driven representations learned using DGMs, for improved surrogate modeling. Second, it is well known in the AI search literature that effective search strategies varies from one problem to another. For example, a recent work on BO over string spaces [42], showed that we need to specialize genetic algorithms for different problems. In contrast, we can use generic search strategies for continuous spaces with LADDER.

**Reinforcement learning (RL)** formulations are also studied to solve this optimization problem [71, 73, 47, 62, 50, 1, 60]. RL is very effective for simulated environments where collection of large amount of data to improve the policy is possible. Unfortunately, this requirement is impractical when we need to perform expensive experiments to collect new training examples.

# 6    Discussion and Limitations

The main contribution of this paper is to improve the performance of latent-space BO methods. However, an alternative approach to solve the problem discussed in this paper is to perform BO directly on the structured space. We evaluate one instance of this approach (BO with Fingerprints kernel) on the chemical design task and results are shown in the below table. We make two observations: **1)** LADDER has better accuracy for a large number of initial BO iterations compared to BO w/ Fingerprints approach; and 2) BO with Fingerprints kernel alone reached slightly better accuracy than LADDER in the end. It should be noted that if we improve the latent space by training deep generative models on larger amounts of unsupervised structures, LADDER will be able to achieve comparable or better accuracy when compared to the BO with Fingerprints kernel approach.

|  | Iteration: 15 | Iteration: 25 | Iteration: 50 | Iteration: 75 | Iteration: 100 |
|---|---|---|---|---|---|
| BO w/ Fingerprints | 2.46 +- 0.32 | 2.80 +- 0.23 | 3.11 +- 0.11 | 3.32 +- 0.23 | 3.36 +- 0.23 |
| LADDER | 2.78 +- 0.26 | 2.93 +- 0.23 | 3.18 +- 0.20 | 3.25 +- 0.18 | 3.25 +- 0.18 |

Moreover, there is a general argument to justify the research direction of combining the strengths of latent space and structured kernels. Given a perfect kernel that captures all the relevant domain-specific characteristics/structure, then the utility for latent space BO methods or LADDER will be limited. However, generic kernels over combinatorial structures (e.g., strings, trees, and graphs) may not always capture the critical domain-specific characteristics. Consider the example of optimizing Metal-Organic Framework (MOF) materials [17] for adsorption property, e.g, storage of gases for hydrogen-powered vehicles. For this problem, we can use graph kernels over the graph representation of MOF materials, but they won't be able to capture some critical characteristics such as pore size, which is important to predict the adsorption property of MOF materials. In such scenarios, domain experts and practitioners can either spend time to engineer a hand-designed kernel (e.g, Fingerprints kernel from the chemical sciences community) or use large amounts of unsupervised structures data to learn a latent representation to capture these domain-specific characteristics in an automated manner. This paper takes the first step to synergistically combine the advantages of latent space and generic/domain-specific kernels to automate the overall BO workflow, which is highly advantageous for scientists and engineers who are the real-users of this technology. We do not claim that LADDER is the optimal algorithm to achieve this goal, but our hope is that LADDER will inspire the BO community to explore this important research direction to develop better algorithms in the future[5].

One potential limitation of LADDER might be selecting the right choice of structured kernel for a given task. Theoretically, the representation power of kernels is studied in terms of properties including universal and characteristic [64]. Satisfying these properties is a basic requirement for selecting a kernel. However, since many known kernels are universal, this criterion might not be enough for all problem settings. Nevertheless, for combinatorial spaces, most of the generic kernels are roughly defined in terms of counting the number of common substructures in a given pair of combinatorial structures [25]. Kernels that capture large sub-structures are generally more powerful in terms of their representational capacity. However, there is a trade-off between representation power and time complexity of kernel computation. For our problem setting of optimizing combinatorial spaces using expensive function evaluations, an instance of small data problem (i.e., with a small number of structure and function evaluation pairs), it is best to select a kernel with the highest representation power and some hyper-parameters that can be optimized in a data-driven manner.

# 7    Summary and Future Work

We introduced a sample-efficient Bayesian optimization (BO) approach for combinatorial spaces called LADDER. The key idea behind LADDER is a Gaussian process based surrogate model that combines the complementary strengths of latent space representation with rich information about decoded outputs using structured kernels. We showed that the BO performance of LADDER is better or similar than state-of-the-art methods and significantly better than the Naïve latent space BO method. Since LADDER's key contribution is in the surrogate model part of the BO procedure, it provides the flexibility to use any acquisition function, which opens up an avenue for various type of extensions including multi-objective [4, 12, 48, 65], multi-fidelity [67, 35, 72], and constrained BO [29, 23].

---

[5]There is concurrent paper [45] at NeurIPS-2021 that leverages epistemic uncertainty of the decoder to guide the optimization process.

**Acknowledgements.** This research is supported in part by NSF grants IIS-1845922, OAC-1910213, and SII-2030159. The authors' would like to thank the anonymous reviewers for their constructive feedback and suggestions to improve the paper.

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
