# OpenReview forum: "Combining Latent Space and Structured Kernels for Bayesian Optimization over Combinatorial Spaces"
_NeurIPS.cc/2021/Conference — NeurIPS 2021 Poster_

### Official Review · Reviewer_KoaP · 2021-07-07

**Rating:** 4
**Confidence:** 5

**Summary:**

This paper proposes a new kernel specifically designed for performing Bayesian optimization in the latent space of deep [latent variable] generative models. The new kernel is formed by taking two existing kernels, one over the latent variable z (e.g. Matern, RBF) and another over the input space x (e.g. a string kernel), then combining them using the Generalized Nystrom Extension. They provide a motivating argument as to why a kernel that uses both x and z might be superior to one that just uses z, and perform experiments which suggest that this kernel improves overall Bayesian optimization performance.

**Limitations And Societal Impact:**

I do not believe that there are potential negative societal impacts of this work.

I think that the authors mentioned some important limitations of the work, but focused heavily on its advantages. I believe that the following limitations should be mentioned or highlighted:

1. Using a predefined structured kernel requires you to choose the *right* kernel, which can be an advantage if you know how to do that but can potentially be a disadvantage. Indeed, this is how the authors explain their poor performance in line 311. However elsewhere you mainly refer to this as an advantage. I think that this point deserves more nuanced treatment.
2. Using a structured kernel means that you can no longer use gradient-based optimization to optimize the GP acquisition function, which is an important advantage of LSBO.

**Main Review:**

## Summary

This paper presents an interesting idea and was exciting to read. The proposed method, LADDER, is a novel combination of several already published methods. They demonstrate promising empirical performance of their method. However, I thought that the paper made a lot of imprecise claims and that the experimental evaluation was missing important baselines.

Because this is a chiefly empirical paper, to recommend acceptance I feel that the authors must show that their combination of methods leads to meaningful performance increases on relevant problems compared to baselines. Given their lack of comparison to important baselines, I was not convinced that their specific combination of methods was responsible for their reported performance increase and think it is plausible that they are instead leveraging advantages of an established method which is known to be superior. Given this I must recommend rejection, but I will be happy to change my mind if presented with new evidence.

## Motivation (sections 1-3.1)

Although I liked the idea of LADDER, to my surprise I disagreed with many statements the authors made when trying to motivate their work. Here are some key examples:

1. Challenge #1 states that naive LSBO "ignores" the decoder, leading to "fundamental disconnect between the modeling process and the objective function" (lines 34, 116). Since the encoder is an approximate inverse of the decoder, even a BO method using just the encoder implicitly has a lot of information from the decoder, just not explicitly. Furthermore, the decoder *is* explicitly used in the naive LSBO procedure, as mentioned in lines 96-97. Therefore I think that the various claims made about "ignoring the decoder" are misleading and should be rephrased. For example, I think that a more moderate and accurate claim would be that "kernels only using z don't explicitly incorporate information about the structure x, only what has been learned by the VAE, which may not have the desired inductive bias to approximate f(x) with a GP.".
2. Challenge #2 states that "the statistical model itself might not be expressive enough to generalize well beyond the training examples from the latent space". While this could happen for simple parametric models, GPs with RBF/similar kernels can approximate any continuous function to arbitrary accuracy [1,2], and therefore modelling failures are unlikely to be caused by the model "not being expressive enough". I agree that a GP may fail to generalize when trained on a small amount of data but I would attribute that to 1) poor choice of kernel/kernel hyperparameters 2) a large search space such that it is easy to find points far away from the training data where almost all ML models will fail to generalize.
3. In line 125 you claim that Figure 2 provides "strong empirical evidence to demonstrate these challenges and our key hypothesis", when in fact it only shows that the predictive accuracy of the posterior mean with your kernel improves faster in the small data regime than the baseline kernel. This figure does not specifically provide evidence that the failure of naive LSBO is due to lack of expressiveness of the statistical model, nor does it show that it is inextricably linked to the kernel not using the decoded x explicitly.

In future versions of this manuscript, I think that the authors should tone down the strength of their claims, and either clearly present their motivation as "conjecture" or try to provide more specific arguments/evidence to support their claims about why naive LSBO doesn't perform well in the low data regime.

## LADDER itself

Using a kernel that incorporates x explicitly is a good idea, and to my knowledge this has not been studied before in the context of BO with VAEs. The structure-coupled kernel that you use does not appear to be novel (after briefly skimming the reference that you cited), but it is certainly novel in this context.

The proposed structure-coupled kernel seems like a reasonable choice, although I think it could be motivated more in the paper. Why was this kernel in particular chosen? Why not a sum kernel like $k(x,x) + l(z,z)$ or a product kernel like $k(x,x)l(z,z)$? A priori these all seem like quite reasonable choices to me. You claim that your specific kernel is "principled" but it's not obvious to me what these principles are...

## Experimental evaluation

### Experiments performed

In general I think that the experiments supported the claims of the paper, although I have some questions and suggestions for improvement.

1. Choice of experiments (logP and arithmetic expression task) was good and in line with prior work (although these tasks are somewhat contrived and I personally hope that the field will move away from them).
2. Setup in section 4.2 is well-described and fairly comprehensive. I noticed 2 small things with your fingerprint kernel: 1) you didn't specify a radius of your fingerprints, which is an important parameter 2) you use the dot product between fingerprints, rather than the jaccard distance which is more standard [3].
3. In "Comparison of surrogate models" (line 271) I had several issues with the experimental setup. Given these I don't think that you can conclude that the LADDER GP is better than the naive LSBO GP (even though I do think it's likely that it actually is better).
    1. Firstly, you don't specify how the 50 random training sets are generated (i.e. random according to what distribution?).
    2. Secondly, MAE is a very unusual evaluation metric for a GP, given that a) marginal log likelihood and not MAE is the GP training objective b) the BO process uses more than just the mean prediction from the GP; it also uses the predictive variance. I therefore think that MLL should be reported instead of MAE, and I'm not sure whether the trend will still hold.
    3. Thirdly, you don't report baselines (e.g. MLL of prior GP, MAE of mean predictor) so it's unclear how significant the decreases in MAE shown are. If for example the variance of the dataset is normalized to be 1, then a MAE of >1 is already quite poor, suggesting that neither model fits the data well.
    4. You don't specify how the GP hyperparameters are chosen for this experiments. You are using ARD kernels in what I presume is >=20 dimensions, so there are >=20 kernel hyperparameters to fit. It is very reasonable to assume that if the hyperparameters are found by MLL maximiation on only ~20 data points that there is a strong risk of overfitting which could explain the poor performance.
4. Naive LSBO vs LADDER experiment (line 285): the methodology seems good here, but you don't seem to mention the number of initial training points used to fit the first GP, which is a very important omission. Nonetheless it does show that LADDER outperforms LSBO.
5. LADDER vs. State-of-the-art (line 296). Again the number of initial training points is not shown: my understanding is that many of the baseline methods will perform better if given more training points, so this number is quite important. The difference for the arithmetic expression task seems quite significant, although it was disappointing that the performance for the chemical design task was not significantly better than the baselines. Thank you for being transparent about this.

### Missing baselines

I think that many important baselines were missing from this work. It is widely known that fingerprint-based methods perform well on a variety of small data tasks. However, you only compare against other latent space optimization methods.

In particular, **I would like to see a comparison against a Tanimoto kernel between the Morgan fingerprints.** I believe that this will be a strong baseline. It would also clarify whether your increased performance is due to using a kernel more similar to the Morgan fingerprint kernel or whether it is truly the combination of the latent space kernel together with the fingerprint kernel. *I think it is quite plausible that the latent space contributes very little to your method and that the reason you have performance increases is because you approach the performance of the Morgan fingerprint GP.* Note that for this baseline, because CMA-ES will no longer be applicable, I believe that your search strategy should be to sample a large number of points from the VAE (e.g. 10000) and choose the one with the highest acquisition function value. This is a key reason why I have decided to recommend rejection, so I would appreciate this point being addressed very directly and explicitly in your rebuttal.

Furthermore, having worked with the ZINC250K dataset, I also think that randomly sampling from the dataset might be competitive with the performance you show in Figures 3-4. It would be nice to see this baseline too. I have less intuition about the arithmetic expression task.

## Related work

The related work mentioned seems sufficiently comprehensive to me.

## Overall review

- Originality: I think the application of a Nystrom extension kernel to latent space Bayesian optimization is original.
- Quality: I had issues with the writing and thought that the paper made too many vague claims, didn't justify important design decisions, and was missing important experiments to convince the reader of the usefulness of their method.
- Clarity: the paper was clear in some parts but very unclear in other parts. I believe that the manuscript could use more polishing
- Significance: the paper makes a contribution to an important problem, but I don't feel convinced that this contribution is large.


# References

[1] https://stats.stackexchange.com/questions/268429/do-gaussian-process-regression-have-the-universal-approximation-property#268884

[2] https://jmlr.csail.mit.edu/papers/volume7/micchelli06a/micchelli06a.pdf

[3] https://en.wikipedia.org/wiki/Jaccard_index#Tanimoto_similarity_and_distance

# Post-rebuttal

Thank you again for your detailed response to my comments. After thinking about it and discussing my concerns with the other reviewers I have decided to keep my score. My primary reasons for this are:

- performance is too comparable with the Tanimoto kernel (a strong baseline). I think that you should come up with an experiment that explicitly demonstrates the difference that the latent space makes here. I think that BO performance is too far removed from the model since it depends on the GP hyperparameters, acquisition function, and a lot of randomness (especially in the low-data setting). Perhaps reporting something like marginal log likelihood would be better, or plotting the predictions made during BO to see why it is doing better.
- Other structured kernels should be explored as baselines
- Structure-coupled kernel could be explained better see my Q5, although this was the least important reason because I trust that you will improve the description in the next version of the manuscript.

**Time Spent Reviewing:**

4

---

> ### Author Response · Authors · 2021-08-10
> **Response to questions/concerns and thanks for the feedback**
>
> Thank you for the detailed feedback with many important questions and pointing out that our paper is interesting with promising empirical performance.  We will clarify the concerns and answer the questions raised in the review comments below.
>
> **Q1**. In particular, I would like to see a comparison against a Tanimoto kernel between the Morgan fingerprints. I believe that this will be a strong baseline ..... This is a key reason why I have decided to recommend rejection, so I would appreciate this point being addressed very directly and explicitly in your rebuttal.
>
> |                                              | Iteration: 15 | Iteration: 25 | Iteration: 50 | Iteration: 75 | Iteration: 100 |
> |----------------------------------------------|---------------|---------------|---------------|---------------|----------------|
> | BO w/ Tanimoto kernel (Fingerprints kernel) | 2.46 +- 0.32  | 2.80 +- 0.23  | 3.11 +- 0.11  | 3.32 +- 0.23  | 3.36 +- 0.23   |
> | LADDER                                       | 2.78 +- 0.26  | 2.93 +- 0.23  | 3.18 +- 0.20  | 3.25 +- 0.18  | 3.25 +- 0.18   |
>
> We present results for the BO w/ Tanimoto kernel (BO with Fingerprints kernel) on the  chemical design task in the table above. The experimental setup is the same as in the main paper. We make two observations:   1) LADDER has better accuracy for a large number of initial BO iterations compared to BO w/ Tanimoto approach; and 2)  BO with Tanimoto (fingerprints) kernel  alone reached slightly better accuracy than LADDER in the end. However, please note that the main contribution of this paper is to improve the performance of latent-space BO methods. Note that if we improve the latent space by training deep generative models on larger amounts of unsupervised structures, LADDER will be able to achieve comparable or better accuracy when compared to the BO w/ Tanimoto approach.
>
> We would like to provide a more general argument to justify the research direction of combining the strengths of latent space and structured kernels. If we have a perfect kernel that captures all the relevant domain-specific characteristics/structure, then there is no utility for latent space BO methods or LADDER. However, generic kernels over combinatorial structures (e.g., strings, trees, and graphs) may not always capture the critical domain-specific characteristics. For example, in optimizing Metal-Organic Framework (MOF) materials for adsorption property, e.g, storage of gases for hydrogen-powered vehicles, we can use graph kernels over the graph representation of MOF materials, but they won’t be able to capture some critical characteristics such as pore size, which is important to predict the adsorption property of the MOF materials. In such scenarios, domain experts and practitioners can either spend time to engineer a hand-designed kernel (e.g, Fingerprints kernel from the chemical sciences community) or use large amounts of unsupervised structures data to learn a latent representation to capture these domain-specific characteristics in an automated manner. Our research takes the first step to synergistically combine the advantages of latent space and generic/domain-specific kernels to automate the overall workflow, which is very advantageous for scientists and engineers who are the real-users of this technology. By no means we claim that LADDER is the optimal algorithm to achieve this goal, but our hope is that LADDER will inspire the BO community to explore this important research direction to uncover better algorithms in the future.
>
> **Q2**. Challenge #1 states that naive LSBO "ignores" the decoder, leading to "fundamental disconnect between the modeling process and the objective function" (lines 34, 116). Since the encoder is an approximate inverse of the decoder, even a BO method using just the encoder implicitly has a lot of information from the decoder, just not explicitly. Furthermore, the decoder is explicitly used in the naive LSBO procedure, as mentioned in lines 96-97. Therefore I think that the various claims made about "ignoring the decoder" are misleading and should be rephrased. For example, I think that a more moderate and accurate claim would be that "kernels only using z don't explicitly incorporate information about the structure x, only what has been learned by the VAE, which may not have the desired inductive bias to approximate f(x) with a GP."
>
> Thank you for your feedback and good suggestions for improved exposition. We agree with your comments and will rephrase the main drawback in the final paper to say that kernels in the naive latent space BO framework don’t explicitly incorporate information about the structure x.
>
>
> **Q3**. Challenge #2 states that "the statistical model itself might not be expressive enough to generalize well beyond the training examples from the latent space". While this could happen for simple parametric models, GPs with RBF/similar kernels can approximate any continuous function to arbitrary accuracy [1,2], and therefore modelling failures are unlikely to be caused by the model "not being expressive enough". I agree that a GP may fail to generalize when trained on a small amount of data but I would attribute that to 1) poor choice of kernel/kernel hyperparameters 2) a large search space such that it is easy to find points far away from the training data where almost all ML models will fail to generalize.
>
> Thank you for your feedback and good suggestions for improved exposition. Regarding challenge #2, we wished to emphasize on the  small data setting (i.e., with a small number of structure and function evaluation pairs in the order of a few 100s) that is common in many real-world science and engineering applications -- the primary concern is to minimize the number of physical lab experiments to perform design optimization. We will incorporate your suggestions to  improve the description of the second challenge by providing the precise qualification.
>
>
> **Q4**. In line 125 you claim that Figure 2 provides "strong empirical evidence to demonstrate these challenges and our key hypothesis", when in fact it only shows that the predictive accuracy of the posterior mean with your kernel improves faster in the small data regime than the baseline kernel. This figure does not specifically provide evidence that the failure of naive LSBO is due to lack of expressiveness of the statistical model, nor does it show that it is inextricably linked to the kernel not using the decoded x explicitly. In future versions of this manuscript, I think that the authors should tone down the strength of their claims, and either clearly present their motivation as "conjecture" or try to provide more specific arguments/evidence to support their claims about why naive LSBO doesn't perform well in the low data regime.
>
> We made this comment based on the following reasoning. There are two major components for a Bayesian optimization algorithm: surrogate model (GP in our case) and acquisition function optimization. Since we kept the acquisition function optimization exactly the same for both Naive LSBO and LADDER, the gains in BO performance can only come from the GP surrogate model which differs only in terms of the kernel. Therefore, we directly compared the regression performance of the two kernels (continuous kernel and structure-coupled kernel) empirically to demonstrate this effect.
>
>
>
>
> **Q5**. The proposed structure-coupled kernel seems like a reasonable choice, although I think it could be motivated more in the paper. Why was this kernel in particular chosen? Why not a sum kernel like  or a product kernel like ? A priori these all seem like quite reasonable choices to me. You claim that your specific kernel is "principled" but it's not obvious to me what these principles are..
>
> You are right that sum and product kernel are valid choices but we didn’t choose them because the acquisition function optimization will require a search over mixed discrete and continuous space which is presumably much harder than continuous search that LADDER employs. We will add appropriate descriptions to justify the choice of structure-coupled kernel over sum and product kernels, and also explicitly state the underlying principle behind the proposed kernel.
>
>
>
> **Q6**. You don't specify how the 50 random training sets are generated (i.e. random according to what distribution?).
>
> They are generated in a uniformly random manner from the dataset.
>
>
> **Q7**. You don't report baselines (e.g. MLL of prior GP, MAE of mean predictor) so it's unclear how significant the decreases in MAE shown are. If for example the variance of the dataset is normalized to be 1, then a MAE of >1 is already quite poor, suggesting that neither model fits the data well.
>
> We don’t normalize the output observations.
>
> **Q8**. You don't specify how the GP hyperparameters are chosen for this experiments. You are using ARD kernels in what I presume is >=20 dimensions, so there are >=20 kernel hyperparameters to fit. It is very reasonable to assume that if the hyperparameters are found by MLL maximization on only ~20 data points that there is a strong risk of overfitting which could explain the poor performance.
>
> Yes, we use L-BFGS based optimization for kernel hyper-parameters.
>
> **Q9**. Naive LSBO vs LADDER experiment (line 285): the methodology seems good here, but you don't seem to mention the number of initial training points used to fit the first GP, which is a very important omission. Nonetheless it does show that LADDER outperforms LSBO.
>
> We use 10 random points (uniformly picked from the dataset) to initialize the GP models.

---

> > ### Comment · Reviewer_KoaP · 2021-08-17
> > **Question about table in rebuttal**
> >
> > Thank you for your response to my concerns. Can you please explain what the numbers in the table are? Is that BO performance, or the MSE of the GP on the training data, or something else?

---

> > > ### Author Response · Authors · 2021-08-18
> > > **Response to Question about table in rebuttal**
> > >
> > > The numbers in the table show BO performance i.e. best objective function value as a function of number of evaluations.

---

> > ### Comment · Reviewer_KoaP · 2021-08-17
> > **Is there no discrete optimization in your method**
> >
> > You state:
> >
> > > You are right that sum and product kernel are valid choices but we didn’t choose them because the acquisition function optimization will require a search over mixed discrete and continuous space which is presumably much harder than continuous search that LADDER employs. We will add appropriate descriptions to justify the choice of structure-coupled kernel over sum and product kernels, and also explicitly state the underlying principle behind the proposed kernel.
> >
> > Do you not need to do any discrete optimization at all to optimize your acquisition function? Doesn't your kernel require the decoded structure term? If so, trying to calculate $\frac{dk}{dz}$ will have a $\frac{d\phi(z)}{dz}$ term, which is undefined because your decoder is not differentiable.
> >
> > Can you clarify how your acquisition function is optimized, and whether it requires any discrete optimization?

---

> > > ### Author Response · Authors · 2021-08-18
> > > **Response to Is there no discrete optimization in your method**
> > >
> > > Thanks! You are right that the acquisition function is not differentiable because of the decoder. Therefore, we use a zeroth-order (but continuous) optimizer CMA-ES which shows good performance.

---

### Official Review · Reviewer_JPaw · 2021-07-09

**Rating:** 7
**Confidence:** 4

**Summary:**

The authors propose LADDER, a BO method based on VAEs, and a new GP kernel that tackles structured spaces.

The contribution of the paper is the introduction of a new GP kernel that takes into account the decoded structures of points in the latent space.

The work is relevant for the BO community, the paper is well written. The experimental section and the structure of the paper can be improved.



**Limitations And Societal Impact:**

In the checklist, the authors say in point 1.c that the potential negative societal impact was discussed in the paper. However, I think they only talk about this in the checklist which is against the writing guidelines. The authors should add this paragraph to the main paper.

Additionally, in point 3.d, while the authors describe the CPU used they don't describe the amount of compute used. This should be added to the main text.


**Main Review:**

In general, the paper is relevant and well-written. I believe that the community will benefit from this type of work and the new kernel will be easy to integrate into existing frameworks.

There are two main negative points:
1) The main criticism of this paper is that it goes into a long description of a problem setting that is fairly well-known at this point. In the last few years, the BO community has seen several papers on Latent Space Optimization (LSO) and an expert reader would be familiar with most of the content in this paper. The new part comes on page 5 from lines 170 to 202, fairly short. While this part is short, it is very relevant for the BO community and this is reflected in my final score.
I would perhaps recommend the authors to go faster to the method and perhaps spend a bit more time describing the following points:
- Why there is a "fundamental disconnect between the modeling process and the objective function", citing the words of the authors. Could you give a simple example to illustrate this point better, where you use the "rich structural information" as a working example?
- Since the paper heavily relies on the Generalized Nystrom extension idea, it'd be useful to use some of the paper space to illustrate that extension a bit more extensively.
- Describe the applications better. The reader has the feeling that it is expected to know these applications already while they are not mainstream applications.

2) The experimental section would be much stronger if the authors used a commonly used Naive LSBO method instead of reimplementing their own. For example, the authors could have used SMAC or Spearmint, which are well-adopted frameworks. This lack of comparison makes Figure 3 much weaker. I strongly recommend the authors add such a comparison in Figure 3. And in that case, the curves in green and orange become irrelevant.
It is also unclear to me how much better the result is in Figure 3 (right). The final performance difference seems pretty small (0.5) and the curves are noisy. So, if you are not an expert on this application you are not able to judge how better those numbers are.

The authors don't mention if they considered retraining the VAE. They mention Tripp et al that do the retraining but I'd be curious to know if they experimented with the retraining and if that gives any improvement.

The authors could make more explicit in the text at their own advantage that they use the same benchmarks as in Tripp et al. with the 2D shape area which would be interesting to add so as to give a more thorough experimental setting. Comparing the results on at least three benchmarks would give more insights into the new method and make the experimental section stronger.



Minor:
- The name LADDER may conflict with the Ladder Variational Autoencoder which is in a similar space to this work:
https://papers.nips.cc/paper/2016/file/6ae07dcb33ec3b7c814df797cbda0f87-Paper.pdf
- Line 98: ".."
- Line 126: it is usually bad writing practice to refer to Figure 2 when you didn't mention Figure 1 yet.
- Line 161-169: it is bad writing practice to describe what GPs are in the middle of the method section. That paragraph should be moved into the background section.
- Line 174: witin -> within
- Equation (1): add comas
- Line 188: "the eigenvector" -> "an eigenvector"
- Figure 2: move the figure to the bottom (or the top) of the page for good writing practice.
- Line 273: there is a grammar error in that sentence.
- Line


**Time Spent Reviewing:**

8 hours

---

> ### Author Response · Authors · 2021-08-10
> **Response to questions/concerns and thanks for the feedback**
>
> Thank you for the nice feedback pointing out that our paper is well-written and relevant for the BO community and can be easily integrated into existing frameworks.  We will clarify the concerns and answer the questions raised in the review comments below.
>
> **Q1**. The main criticism of this paper is that it goes into a long description of a problem setting that is fairly well-known at this point. In the last few years, the BO community has seen several papers on Latent Space Optimization (LSO) and an expert reader would be familiar with most of the content in this paper. The new part comes on page 5 from lines 170 to 202, fairly short. While this part is short, it is very relevant for the BO community and this is reflected in my final score. I would perhaps recommend the authors to go faster to the method and perhaps spend a bit more time describing the following points:
>
> Our intention behind the exposition choice was to make the paper accessible to a broader range of audience including general BO researchers/practitioners and not necessarily experts in combinatorial BO and latent space BO methods. We will try to incorporate some of your feedback in the final paper.
>
> **Q2**. Why there is a "fundamental disconnect between the modeling process and the objective function", citing the words of the authors. Could you give a simple example to illustrate this point better, where you use the "rich structural information" as a working example?
>
> For example, let Z is the latent space and X is the combinatorial space. Commonly, latent space BO approaches employ GP models over Z space. However, the objective function is defined over the X space. Each z point in the latent space is associated with an objective function value by first decoding it into an x point in the combinatorial space and evaluating the objective function on x.
>
> One of the reviewers explained this point much more clearly as “kernels only using z don't explicitly incorporate information about the structure x, only what has been learned by the VAE, which may not have the desired inductive bias to approximate f(x) with a GP." We will rephrase this explanation in the final paper.
>
> **Q3**. Since the paper heavily relies on the Generalized Nystrom extension idea, it'd be useful to use some of the paper space to illustrate that extension a bit more extensively.
>
> We will add a more detailed description of the Generalized Nystrom extension idea in the final paper.
>
> **Q4**. Describe the applications better. The reader has the feeling that it is expected to know these applications already while they are not mainstream applications.
>
> We will update the description of all the applications in the final paper.
>
>
> **Q5**. The experimental section would be much stronger if the authors used a commonly used Naive LSBO method instead of reimplementing their own. For example, the authors could have used SMAC or Spearmint, which are well-adopted frameworks. This lack of comparison makes Figure 3 much weaker. I strongly recommend the authors add such a comparison in Figure 3. And in that case, the curves in green and orange become irrelevant. It is also unclear to me how much better the result is in Figure 3 (right). The final performance difference seems pretty small (0.5) and the curves are noisy. So, if you are not an expert on this application you are not able to judge how better those numbers are.
>
> First, we would like to clarify that the code for the proposed LADDER approach was built on top of a publicly available code (https://github.com/cambridge-mlg/weighted-retraining; Tripp et al., NeurIPS-2020). We can add the results with SMAC (Random Forest surrogate model) or Spearmint in the final paper, but we don’t believe the comparison results will change.  Second, arithmetic expressions and chemical design tasks are two commonly employed benchmarks in all prior work on latent space BO papers. We clearly show relative improvement with LADDER over the latent space BO framework.
>
> **Q6**. The authors don't mention if they considered retraining the VAE. They mention Tripp et al that do the retraining but I'd be curious to know if they experimented with the retraining and if that gives any improvement.
>
> Yes, we performed some preliminary experiments with the retraining method of Tripp et al.. The results slightly improved over the Naive latent space BO method, but were slightly worse than LADDER. The key reason being the small-data problem setting (i.e., with a small number of structure and function evaluation pairs in the order of a few 100s) considered in this paper.
>
> **Q7**. The authors could make more explicit in the text at their own advantage that they use the same benchmarks as in Tripp et al. with the 2D shape area which would be interesting to add so as to give a more thorough experimental setting. Comparing the results on at least three benchmarks would give more insights into the new method and make the experimental section stronger.
>
> Thank you for these suggestions. We will work on them as part of our immediate future work.

---

### Official Review · Reviewer_iwAZ · 2021-07-16

**Rating:** 4
**Confidence:** 4

**Summary:**

This paper proposes a Bayesian optimization method to solve a combinatorial optimization problem by combining a latent space and structured kernels. Because optimizing an unknown function on a combinatorial space via Bayesian optimization is intractable, a problem is converted to a problem of optimizing on a continuous space (i.e., a latent space). The authors suggest a principled approach, dubbed LADDER. The main idea is to integrate structural information from decoded structures. Finally, the experimental results show that the proposed method outperforms other baseline methods.

**Ethical Concerns:**

I do not have any ethical concerns.

**Limitations And Societal Impact:**

I do not have any comments on limitations and societal impact. Please see "Main Review" for the detailed reviews.

**Main Review:**

Combinatorial optimization is one of the most important problems in many science and engineering fields. This paper solves such a problem using Bayesian optimization, and the authors suggest a new approach to solving an optimization problem on a latent space. However, I have some concerns on novelty and numerical experiments. Please see the comments described below and provide a response for those comments in the rebuttal.

Pros

+ The topic solved in this work is very interesting.
+ The paper is well-written and well-organized.

Cons

- The novelty of this work is limited, since similar studies [1, 4] have been proposed recently. They should be discussed in the paper.
- I think some well-known baselines such as [2, 3, 5] are not included in the experiments. In my experience, those baselines might show robust performance in combinatorial optimization.
- Real-world experiments are needed.

Detailed Comments

1. I would like to ask if the authors have a specific reason that does not include an approach based on Bayesian optimization as a baseline.
2. How do you determine a search space for an acquisition function, i.e., $\mathcal{Z}$? If this space is not well-defined, this approach might not work. Or, if there is a method to restrict an embedded vector into a compact and convex space, please describe in the rebuttal.
3. How do you train a pair of encoder and decoder? Even though it is trained via unsupervised manner, it should be described in detail.
4. Can you guarantee that a decoder always creates a valid example on combinatorial space? For example, if a chemical compound that is determined by a decoder might be invalid or impossible to synthesize, how did you cope with it?

Minor issues

* In Line 9, *doesn't* should be *does not*.
* In Line 48, *Second, allows* should be *Second, it allows*. Please use a formal expression.
* In Line 112, an equal symbol $=$ seems like not a mathematical font. Please check it.

[1] R. Moriconi, M. P. Deisenroth, and K. Kumar. High-dimensional Bayesian optimization using low-dimensional feature spaces. Machine Learning, 109 (9):1925–1943, 2020.

[2] R. Baptista and M. Poloczek. Bayesian optimization of combinatorial structures. In Proceedings of the International Conference on Machine Learning (ICML), pages 462–471, Stockholm, Sweden, 2018.

[3] C. Oh, J. Tomczak, E. Gavves, and M. Welling. Combinatorial Bayesian optimization using the graph cartesian product. In Advances in Neural Information Processing Systems (NeurIPS), volume 32, pages 2914–2924, Vancouver, British Columbia, Canada, 2019.

[4] T. Iwata and T. Otsuka. Efficient Transfer Bayesian Optimization with Auxiliary Information. arXiv preprint arXiv:1909.07670, 2019.

[5] H. B. Moss, D. Beck, J. Gonzalez, D. S. Leslie, and P. Rayson. BOSS: Bayesian Optimization over String Spaces. In Advances in Neural Information Processing Systems (NeurIPS), volume 33, Virtual, 2020.

**Time Spent Reviewing:**

5

---

> ### Author Response · Authors · 2021-08-10
> **Response to questions/concerns and thanks for the feedback**
>
> Thank you for the thorough feedback and pointing out that our paper is well-written and organized and tackles an interesting problem. We will clarify the concerns and answer the questions raised in the review comments below.
>
> **Q1**. The novelty of this work is limited, since similar studies [1, 4] have been proposed recently. They should be discussed in the paper.
>
> Thank you for the references. Both these papers are unrelated to the proposed LADDER approach. They use some form of auxiliary information. [1] is an approach for high-dimensional BO with the assumption of ‘low-effective’ dimensionality and uses a manifold multi-output GP. [4] is an approach for transfer BO setting which finds the maximum of an expensive-to evaluate black-box function by using data on related optimization tasks. In this paper, we consider a single-task setting where no other similar task is available. We will still add a discussion about these two methods in the final paper.
>
> **Q2**. I think some well-known baselines such as [2, 3, 5] are not included in the experiments. In my experience, those baselines might show robust performance in combinatorial optimization. I would like to ask if the authors have a specific reason that does not include an approach based on Bayesian optimization as a baseline.
>
> The main contribution of this paper is to improve the naive latent space BO approach. We agree that the baselines are related but most of them are applicable in specialized settings. [2] is applicable for fixed size binary structures alone, [3] is applicable for fixed size categorical/binary inputs, and [5] is applicable for string spaces. For the two real-world tasks (arithmetic expressions and chemical design) commonly used for the evaluation of latent space BO approaches, [1] and [2] cannot be used due to varying size input structures, and [5] can be used only for structures that can be represented as strings as in arithmetic expressions task, but the chemical design task involves graph structures. We will add results for BO with the string kernel alone for the arithmetic expressions task in the final paper. Moreover, these methods [2,3,5] require specialized acquisition function optimization routines (e.g., genetic algorithms) whereas our approach allows acquisition function optimization over continuous spaces.
>
> **Q3**. Real-world experiments are needed.
>
> The two benchmark tasks (arithmetic expressions and chemical design) considered in this paper are the commonly used ones for the evaluation of latent space BO approaches [Reference 1, 2, 3]. Since the proposed LADDER method is designed to improve the latent space BO framework, we used the same benchmarks. In fact, the chemical design task is a proper real-world optimization problem.
>
> [Reference 1] Tripp, A., Daxberger, E., & Hernández-Lobato, J. M. (2020). Sample-efficient optimization in the latent space of deep generative models via weighted retraining. Advances in Neural Information Processing Systems, 33.
>
> [Reference 2] Jin, W., Barzilay, R., & Jaakkola, T. (2018, July). Junction tree variational autoencoder for molecular graph generation. In International conference on machine learning (pp. 2323-2332). PMLR.
>
> [Reference 3] Kusner, M. J., Paige, B., & Hernández-Lobato, J. M. (2017, July). Grammar variational autoencoder. In International Conference on Machine Learning (pp. 1945-1954). PMLR.
>
>
> **Q4**. How do you determine a search space for an acquisition function, i.e., ? If this space is not well-defined, this approach might not work. Or, if there is a method to restrict an embedded vector into a compact and convex space, please describe in the rebuttal
>
> Determining the correct search space is an important research problem in the general BO area, which deserves a separate investigation in its own right. However, in our paper, we assume a  rectangular closed boundary [-a, a]^D which is a common approach in many BO papers. Specifically, we used the same continuous latent space as prior work [Reference 1] and built the code of LADDER on top of their publicly available code.
>
> [Reference 1] Tripp, A., Daxberger, E., & Hernández-Lobato, J. M. (2020). Sample-efficient optimization in the latent space of deep generative models via weighted retraining. Advances in Neural Information Processing Systems, 33. Code: https://github.com/cambridge-mlg/weighted-retraining
>
>
> **Q5**. How do you train a pair of encoder and decoder? Even though it is trained via unsupervised manner, it should be described in detail.
>
> We used publicly available pre-trained models for our purpose which shows the flexibility of our approach. We will add more description about  training deep generative models in the final paper.
>
> **Q6**. Can you guarantee that a decoder always creates a valid example on combinatorial space? For example, if a chemical compound that is determined by a decoder might be invalid or impossible to synthesize, how did you cope with it?
>
> We try decoding the latent space input a few times (upto 10 times in our experiments) in order to get a valid decoding. If we don’t get a valid decoding after 10 tries, we discard the input. This happens very rarely in the arithmetic expressions dataset and never occurs for the chemical design dataset.

---

### Official Review · Reviewer_mmhZ · 2021-07-18

**Rating:** 6
**Confidence:** 4

**Summary:**

This paper proposes to improve Bayesian optimization over combinatorial space by introducing a structure-coupled kernel for the Bayesian surrogate model for BO. The structure-coupled kernel takes not only the embedded location in the latent space but also the corresponding representations in the combinatorial space. It uses a stationary kernel as well as a kernel for structured space such as string kernel and fingerprints kernel and combines them using a kernel construction based on generalized Nystrom extension. The proposed method is compared with the previous latent space based BO method as well as other state-of-the-art BO methods on the Arithmetic expression and chemical design datasets.

**Limitations And Societal Impact:**

The authors have discussed the limitations of the proposed method.


**Main Review:**

This paper tries to tackle a limitation of the current latent space based BO approach for searching combinatorial space. It is a known issue that the surrogate model in BO does not take account of the information of the original structured representations. The proposed method feeds the information of the structured representation into the surrogate model via a structure-coupled kernel.

The paper is easy to follow, although sections 1-3 are a bit repetitive. The proposed method is quite interesting and the experimental results are promising. This work could be potentially a good contribution to the BO community.

I have a few reservations regarding the proposed method and am happy to change my rating according to the authors’ feedback.

1. Could the author provide some intuition about how the structure-coupled kernel behaves? It is easy to see that for the training points the covariance matrix equals L as K K^{-1} L K^{-1} K = L, but what about other data points?

2. How does the proposed method compare with only using a structured kernel? In other words, what is the benefit of using the covariance matrix L from the embedded locations?

3. The proposed method is computationally very expensive when there is lots of data due to the cubic computational complexity of GP. On the other hand, the latent space embedding model like the mentioned VAEs needs a lot of data in order to achieve a good embedding. In the experiments, the sizes of the training sets are all very small (up to 100). With this data size, VAE is known to perform badly. (i) Is latent space embedding helpful in this scenario? What would be the performance of only using a structured kernel such as a string kernel? (ii) If we increase the training set size to a few thousands, how does the proposed method compare with the pure latent space based method?

Missing Reference:
“Structured variationally auto-encoded optimization”, X Lu, J Gonzalez, Z Dai, ND Lawrence, ICML, 2018.


**Time Spent Reviewing:**

2

---

> ### Author Response · Authors · 2021-08-10
> **Response to questions/concerns and thanks for the feedback**
>
> Thank you for the thorough feedback and pointing out that our paper is interesting with promising experimental results and a good contribution to the BO community. We will clarify the concerns and answer the questions raised in the review comments below.
>
> **Q1**. Could the author provide some intuition about how the structure-coupled kernel behaves? It is easy to see that for the training points the covariance matrix equals L as K K^{-1} L K^{-1} K = L, but what about other data points?
>
> As the name suggests, one way to interpret  the proposed structure-coupled kernel idea is as an extension of the more commonly known Nystrom method, which  allows us to extrapolate eigenfunctions known at a fixed set of points. This interpretation can be seen by setting K = L in equation 4 which gives the exact Nystrom extension equation. For points not in the training set, the structured kernel k acts like a smooth extrapolating kernel (For Nystrom, the same kernel acts as both extrapolating and base kernel).
>
> **Q2**. How does the proposed method compare with only using a structured kernel? In other words, what is the benefit of using the covariance matrix L from the embedded locations?
>
> |                            | Iteration: 15 | Iteration: 25 | Iteration: 50 | Iteration: 75 | Iteration: 100 |
> |----------------------------|---------------|---------------|---------------|---------------|----------------|
> | BO w/  Fingerprints kernel | 2.46 +- 0.32  | 2.80 +- 0.23  | 3.11 +- 0.11  | 3.32 +- 0.23  | 3.36 +- 0.23   |
> | LADDER                     | 2.78 +- 0.26  | 2.93 +- 0.23  | 3.18 +- 0.20  | 3.25 +- 0.18  | 3.25 +- 0.18   |
>
> We present results for the ‘BO with only structured kernel approach’ (BO with Fingerprints kernel) on the  chemical design task in the table above. The experimental setup is the same as in the main paper. We make two observations:   1) LADDER has better accuracy for a large number of initial BO iterations compared to the 'BO with only structured kernel' approach; and 2)  BO with fingerprints kernel  alone reached slightly better accuracy than LADDER in the end. However, please note that the main contribution of this paper is to improve the performance of latent-space BO methods. Note that if we improve the latent space by training deep generative models on larger amounts of unsupervised structures, LADDER will be able to achieve comparable or better accuracy when compared to the 'BO with only structured kernel' approach.
>
> Providing similar results for the arithmetic expressions task requires significantly more time due to the computationally expensive string kernel. We will add the corresponding results in the final paper.
>
> We would like to provide a more general argument to justify the research direction of combining the strengths of latent space and structured kernels. If we have a perfect kernel that captures all the relevant domain-specific characteristics/structure, then there is no utility for latent space BO methods or LADDER. However, generic kernels over combinatorial structures (e.g., strings, trees, and graphs) may not always capture the critical domain-specific characteristics. For example, in optimizing Metal-Organic Framework (MOF) materials for adsorption property, e.g, storage of gases for hydrogen-powered vehicles, we can use graph kernels over the graph representation of MOF materials, but they won’t be able to capture some critical characteristics such as pore size, which is important to predict the adsorption property of the MOF materials. In such scenarios, domain experts and practitioners can either spend time to engineer a hand-designed kernel (e.g, Fingerprints kernel from the chemical sciences community) or use large amounts of unsupervised structures data to learn a latent representation to capture these domain-specific characteristics in an automated manner. Our research takes the first step to synergistically combine the advantages of latent space and generic/domain-specific kernels to automate the overall workflow, which is very advantageous for scientists and engineers who are the real-users of this technology. By no means we claim that LADDER is the optimal algorithm to achieve this goal, but our hope is that LADDER will inspire the BO community to explore this important research direction to uncover better algorithms in the future.
>
>
> **Q3**. The proposed method is computationally very expensive when there is lots of data due to the cubic computational complexity of GP. On the other hand, the latent space embedding model like the mentioned VAEs needs a lot of data in order to achieve a good embedding. In the experiments, the sizes of the training sets are all very small (up to 100). With this data size, VAE is known to perform badly. (i) Is latent space embedding helpful in this scenario? What would be the performance of only using a structured kernel such as a string kernel? (ii) If we increase the training set size to a few thousands, how does the proposed method compare with the pure latent space based method?
>
> There seems to be some confusion.  First, the latent space embedding is pre-trained on a large unsupervised dataset (50K for arithmetic expressions task and 250K for chemical design task). Second, we are in a small-data setting  (i.e., with a small number of structure and function evaluation pairs in the order of a few 100s). This means training a GP based surrogate model is not expensive. Importantly, in real-world scientific applications, our primary goal is to minimize the number of real physical lab experiments and computational time for training and selecting structures for evaluation is a secondary concern.  Third, LADDER is designed with small-data problem setting in mind as mentioned in the problem setup. Indeed, we believe that the pure latent space based approach will also improve with increasing data because the GP model will fit better on the latent space (which is high-dimensional) with a larger supervised dataset. In fact, prior work [Reference 1] has already shown evidence for this result.
>
> [Reference 1] Tripp, A., Daxberger, E., & Hernández-Lobato, J. M. (2020). Sample-efficient optimization in the latent space of deep generative models via weighted retraining. Advances in Neural Information Processing Systems, 33.
>
> **Q4** Missing Reference: “Structured variationally auto-encoded optimization”, X Lu, J Gonzalez, Z Dai, ND Lawrence, ICML, 2018.
>
> Thank you for the reference. We will add it in the final paper.

---

> > ### Comment · Reviewer_mmhZ · 2021-08-24
> > **Reply**
> >
> > Thanks for the authors providing additional experiment results and extensive explanation.
> >
> > Overall, I feel that it is a good paper. The additional experiment shows the effectiveness of the structured kernel. IMO, the limitation of the paper is that it could have provided better theoretical and/or emperical analyses of the proposed kernel construction. The movitation for combining a structured kernel and a latent space based kernel makes sense, but the paper does not provide good reasoning about why the generalized Nystrom formulation is a good way of achieving the combination. Therefore, I will keep my current rating.

---

### Decision · Program_Chairs · 2021-09-28

**Decision:**

Accept (Poster)

**Comment:**

We thank the authors for the additional clarifications provided in the rebuttal. All reviewers agreed that this work made interesting contributions and is relevant to the community. However, the consensus was that it missed important related work and baselines. In addition to the references indicated by the reviewers, the authors also missed [XYZ] where an explicit link was made to string kernels. Also, additional experiments conducted by authors showed that the improvement were marginal compared to missing comparable baselines. The fact that the proposed approach (namely, the combination of a structured kernel and a latent space based on the generalized Nystrom) lacked motivation as indicated by several reviewers suggests this work is not ready for publication yet.


[XYZ] R. Jenatton, et al. (2017): Bayesian Optimization with Tree-structured Dependencies. ICML 34.

**Consistency Experiment:**

NeurIPS has a long history of experimentation. In 2014, NeurIPS ran an experiment in which 10% of submissions were reviewed by two independent committees to quantify the randomness in the review process. This year, we repeated a variant of this experiment to see how the quality of the review process has changed over time.  This paper was part of the experiment and was therefore assigned to two committees (consisting of reviewers, an Area Chair, and a Senior Area Chair) that reached independent decisions.  If both committees made the same recommendation, this recommendation was followed. If a single committee recommended acceptance, the paper was accepted (with the exception of a few cases in which the other committee identified what we considered a fatal flaw, e.g., an error in a key result).

This copy’s committee reached the following decision: **Reject**

The other committee assigned to the paper recommended **Accept (Poster)**.  You can find the other set of reviews, along with any follow up discussion with the authors here:
https://openreview.net/forum?id=fxHzZlo4dxe